# Accelerated remyelination during inflammatory demyelination prevents axonal loss and improves functional recovery

Feng Mei[1,2†], Klaus Lehmann-Horn[1†], Yun-An A Shen[1], Kelsey A Rankin[1], Karin J Stebbins[3], Daniel S Lorrain[3], Kara Pekarek[1], Sharon A Sagan[1], Lan Xiao[2], Cory Teuscher[4], H-Christian von Büdingen[1], Jürgen Wess[5], J Josh Lawrence[6], Ari J Green[1], Stephen PJ Fancy[1,7], Scott S Zamvil[1], Jonah R Chan[1*]

[1]Department of Neurology, University of California, San Francisco, San Francisco, United States; [2]Department of Histology and Embryology, Chongqing Key Laboratory of Neurobiology, Third Military Medical University, Chongqing, China; [3]Inception Sciences, San Diego, United States; [4]Department of Medicine, Immunobiology Program, University of Vermont, Burlington, United States; [5]Molecular Signaling Section, Laboratory of Bioorganic Chemistry, National Institute of Diabetes and Digestive and Kidney Diseases, National Institutes of Health, Bethesda, United States; [6]Department of Pharmacology and Neuroscience, Texas Tech University Health Sciences Center, School of Medicine, Lubbock, United States; [7]Department of Pediatrics, University of California, San Francisco, San Francisco, United States

*For correspondence: jonah.chan@ucsf.edu

†These authors contributed equally to this work

Competing interests: The authors declare that no competing interests exist.

**Abstract** Demyelination in MS disrupts nerve signals and contributes to axon degeneration. While remyelination promises to restore lost function, it remains unclear whether remyelination will prevent axonal loss. Inflammatory demyelination is accompanied by significant neuronal loss in the experimental autoimmune encephalomyelitis (EAE) mouse model and evidence for remyelination in this model is complicated by ongoing inflammation, degeneration and possible remyelination. Demonstrating the functional significance of remyelination necessitates selectively altering the timing of remyelination relative to inflammation and degeneration. We demonstrate accelerated remyelination after EAE induction by direct lineage analysis and hypothesize that newly formed myelin remains stable at the height of inflammation due in part to the absence of MOG expression in immature myelin. Oligodendroglial-specific genetic ablation of the M1 muscarinic receptor, a potent negative regulator of oligodendrocyte differentiation and myelination, results in accelerated remyelination, preventing axonal loss and improving functional recovery. Together our findings demonstrate that accelerated remyelination supports axonal integrity and neuronal function after inflammatory demyelination.

## Introduction

Demyelination in multiple sclerosis (MS) results from an aberrant immune response that targets the myelin sheath in the central nervous system (CNS) (*Franklin, 2002*). Over the course of the disease, demyelinated axons undergo irreversible degeneration, which correlates with progression and results in permanent disability—events that are pathological hallmarks of progressive MS

(*Trapp et al., 1998*; *Bitsch et al., 2000*; *Bjartmar et al., 2003*; *Franklin et al., 2012*; *Kuhlmann et al., 2002*; *Lovas et al., 2000*; *Tallantyre et al., 2009*; *Trapp and Stys, 2009*; *Singh et al., 2013*; *Sorbara et al., 2014*). To date, there are no therapeutic interventions that directly prevent neuronal degeneration, especially as the precise mechanisms underlying degeneration remain undetermined (*Bjartmar and Trapp, 2003*; *Simons et al., 2014*). As a crucial accessory to the functional nerve-fiber unit, the myelin sheath provides multiple-layers of concentric membrane, which act to maximize the speed and reduce the energy demands of action potentials. The importance of myelin in the developing CNS is further illustrated by recent findings demonstrating that oligodendrocytes provide critical metabolic support to neurons (*Fünfschilling et al., 2012*; *Lappe-Siefke et al., 2003*; *Lee et al., 2012*). However, while it seems intuitive that remyelination plays a crucial role in preserving axonal integrity and restoring neuronal function in an inflammatory condition like MS, direct evidence is lacking. Remyelination is an inefficient process in MS, which is likely due to an inhibitory microenvironment that prevents oligodendrocyte precursor cells (OPCs) from terminally differentiating into myelinating oligodendrocytes (*Chang et al., 2002*; *Huang and Franklin, 2011*; *Kuhlmann et al., 2008*; *Wolswijk, 1998*). Additionally, it is thought that the microenvironment may also modify the epigenetic regulation of OPCs to alter transcriptional programs necessary for differentiation during development and after demyelination (*Liu et al., 2016a*, *2016b*; *Moyon et al., 2016*; *Huynh et al., 2014*). In order to promote myelin repair in MS, recent high-throughput screening efforts have identified a number of myelin regenerative compounds (*Deshmukh et al., 2013*; *Mei et al., 2014*; *Najm et al., 2015*; *Samanta et al., 2015*). Interestingly, some of these compounds promote precocious differentiation of oligodendrocytes and attenuate clinical symptoms in EAE, an inflammatory demyelinating animal model (*Liu et al., 2016b*; *Mei et al., 2016*; *Deshmukh et al., 2013*; *Najm et al., 2015*; *Samanta et al., 2015*). The mechanisms underlying this phenomenon remain unclear, as active inflammation dynamically results in coincident demyelination, axonal degeneration and remyelination (*Miller and Karpus, 2007*; *Nikić et al., 2011*; *Ransohoff, 2012*; *Stanley and Pender, 1991*). Demonstrating the functional significance of remyelination necessitates uncoupling the immunological response and degenerative process from oligodendrocyte-mediated repair. Here, we uncouple remyelination from inflammation and degeneration by genetically targeting cell-specific deletion of the M1 muscarinic receptor (Chrm1) in OPCs to demonstrate that accelerated remyelination is sufficient to reduce the severity of EAE clinical scores, attributable in part to preserving axonal integrity.

## Results

### Demyelination, axonal loss and remyelination in EAE

To determine whether EAE is a model suitable for investigating remyelination and axonal protection, we set out to characterize the timing of demyelination, axonal loss and the possibility of remyelination throughout the course of EAE. We induced EAE using the MOG peptide (MOG$_{35-55}$) with mice on the C57BL/6 genetic background (*Mendel et al., 1995*). The mice developed chronic paralysis and reached a peak clinical score approximately 14 days post immunization (PI), which was maintained until 30 days PI. To characterize demyelination of the spinal cords during EAE, we employed osmium tetroxide (OsO4; a lipophilic transition metal that efficiently stains myelin) to examine the integrity of the myelin along the white matter tracts of the spinal cord (*Figure 1a–c*). Demyelinating lesions were detected in the white matter tracts of the lumbar spinal cord at the peak of EAE severity (*Figure 1b*) followed by extensive and global demyelination at the late stage of EAE (*Figure 1c*). The widespread damage to myelin was also visible by immunostaining for MOG, a myelin-specific protein, on cross sections of spinal cord (*Figure 1d–f*). Notably, neurofilament (NF) staining displayed numerous demyelinated axons (MOG-negative areas) at the peak of clinical severity (*Figure 1e*), and significant loss of axons by NF immunoreactivity at the late stage of EAE–indicating significant neuronal degeneration (*Figure 1f*). By transmission electron microscopy (EM) (*Figure 1g–i*), remyelinated axons were sparsely identified as thinly myelinated axons in the late stage of EAE spinal cords, which were accompanied with a large number of demyelinated and possibly degenerating axons with diffuse cytoplasmic staining (*Figure 1h,i*). These results indicate that the EAE model is complicated by significant inflammation, demyelination, axonal degeneration and sparse remyelination. These findings suggest that standard EAE may not be an ideal model to investigate the

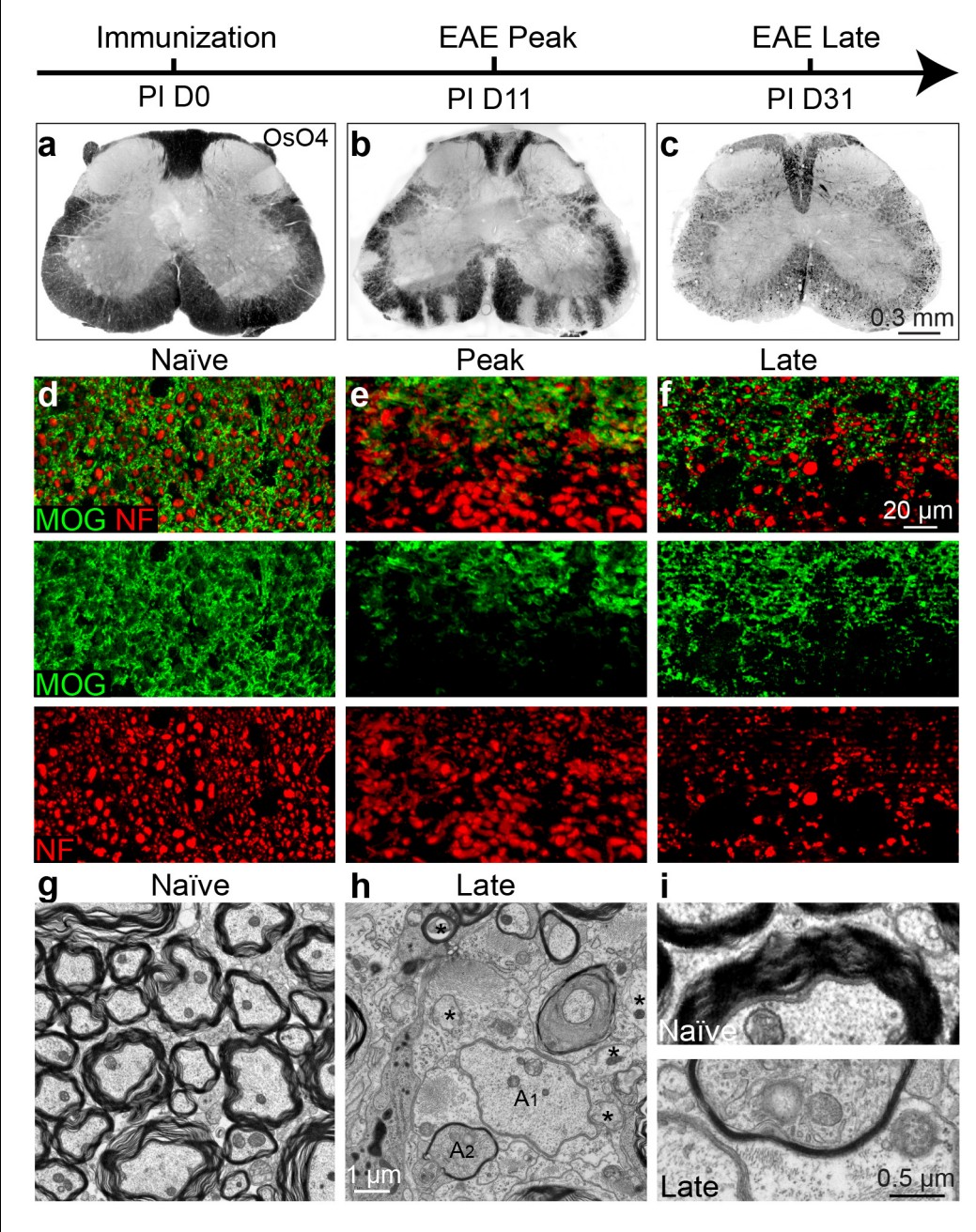

**Figure 1.** Axonal degeneration and remyelination after inflammatory demyelination in EAE. Representative spinal cord sections were stained for OsO4 (black) showing myelin in the naïve (a), and EAE mice at the peak (b) and late stages (c) of EAE. Demyelination is indicated as unstained regions in the white matter tracts. Representative cross sections of spinal cord were immunostained for MOG (green) and NF (red) (d–f) showing myelin and axons in the naïve (d) and the EAE mice at the peak (e) and at late stages of disease (f). (g) Representative transmission electron microscopic images showing normally myelinated axons in naïve mice. h, (i) Representative transmission electron microscopic images showing demyelinated (A1), remyelinated (A2) and possibly degenerating axons (*; diffuse cytoplasmic staining) at the late stage of EAE in spinal cord white matter.

possible relationship between remyelination and axonal integrity. However, as inflammatory demyelination is highly relevant to MS, we decided to pursue approaches that may uncouple the immunological response and degenerative processes from the reparative impact by oligodendrocytes.

## Small molecule approaches do not definitively demonstrate remyelination in EAE

Recently, multiple groups have performed high-throughput screens for identifying therapeutic compounds for remyelination (*Deshmukh et al., 2013*; *Mei et al., 2014*; *Najm et al., 2015*; *Mei et al., 2016*), and have applied some of these compounds in different EAE models with limited success. Among them is clemastine, recently identified in a cluster of anti-muscarinic compounds for remyelination (*Deshmukh et al., 2013*; *Mei et al., 2014*), that also significantly promotes myelination (*Liu et al., 2016b*). To test the efficacy of clemastine in MOG-induced EAE, we initiated daily oral-treatment of clemastine in mice from the onset of immunization (PI 0D). Prophylactic treatment of clemastine significantly decreased the clinical severity of EAE at the peak and throughout the chronic phase of the disease (*Figure 2a*). Remyelination and axon loss were assessed by FluoroMyelin detection and immunostaining for NF in the lumbar spinal cords (*Figure 2b,c*). Clemastine treatment significantly preserved myelin staining intensity and prevented axon loss as assessed by NF immunostaining in the white matter tracts at the late stage of EAE (*Figure 2b,c*). To examine the infiltration of T-cells, macrophages and the activation of microglia, we immunostained for CD3 (T-cells) and Iba1 (macrophages and microglia) and quantified the density of these cells in demyelinated lesions at the late stage of EAE (*Figure 2d*). We did not find a statistical difference in the density of CD3 or Iba1 positive cells after treatment with clemastine, suggesting that the mechanism of action does not alter the inflammatory response at the level of cellular infiltration (*Figure 2e*). However, as histology provides a static image of one time point, it is impossible to conclusively determine whether clemastine, or any other small molecule for that matter, contributes to the attenuation of the clinical score in EAE by solely accelerating remyelination. With that said, the possibility remains that clemastine could still modulate other aspects of inflammation, provide some sort of axonal support, or even stabilize the myelin sheath thereby preventing further damage. It is also important to note that all of these potential effects are not mutually exclusive and may each contribute to the attenuation of the clinical score.

To determine if remyelination is significantly accelerated by clemastine in the EAE model, we took advantage of a genetic approach to examine new myelin sheath generation (remyelination) by using the *Cspg4*-CreErt; *Mapt*-mGFP line in EAE (*Figure 3a*) (*Hippenmeyer et al., 2005*; *Young et al., 2013*; *Etxeberria et al., 2016*). In this line, recombination is induced by tamoxifen only in NG2 cells (OPCs) and newly formed myelin (remyelination) can be visualized by expression of the membrane-associated isoform of GFP (mGFP), as only mature oligodendrocytes express tau (*Figure 3a*). We induced recombination with a single dose of tamoxifen two days prior to immunization and clemastine treatment. GFP-positive myelin was absent in the vehicle EAE mouse at the peak of disease (*Figure 3—figure supplement 1a*), while a number of GFP positive myelin sheaths were visualized as 'rings' surrounding demyelinated axons (NF+/MOG-) after clemastine treatment (*Figure 3b*). These results directly indicate that newly formed myelin (remyelinated axons) can be initiated and identified as early as the peak of disease with myelin regenerative compounds. However, if remyelination is occurring at the height of inflammation, how is the new myelin stabilized in this highly toxic milieu? We hypothesize that as remyelination is accelerated in the MOG-induced EAE model, the new myelin does not yet express the MOG antigen and would be excluded from MOG-induced inflammatory demyelination. This is clearly evident upon the detection of a number of axons surrounded by thin GFP membranes that are still yet negative for MOG expression (*Figure 3c*). We also examined the recombination efficiency of the NG2-CreErt; tau-mGFP mice, as we did not detect any GFPpositive myelin sheaths in the vehicle control mice. As newly formed oligodendrocytes are not generated in high frequency in adult mice, we examined 14-day old mice to determine recombination efficiency by administration of tamoxifen at postnatal day 8 and then examining recombination 6 days later. By examining cortical regions where myelination is sparse at postnatal day 14, we quantified the number of GFP positive, CC1 positive oligodendrocytes and calculated a recombination efficiency of approximately 30% of the total number of oligodendrocytes (*Figure 3—figure supplement 1b*). It is also important to note that we did not identify GFP positive cells in the cortex that were not CC1 positive, suggesting that GFP expression was specific to oligodendrocytes.

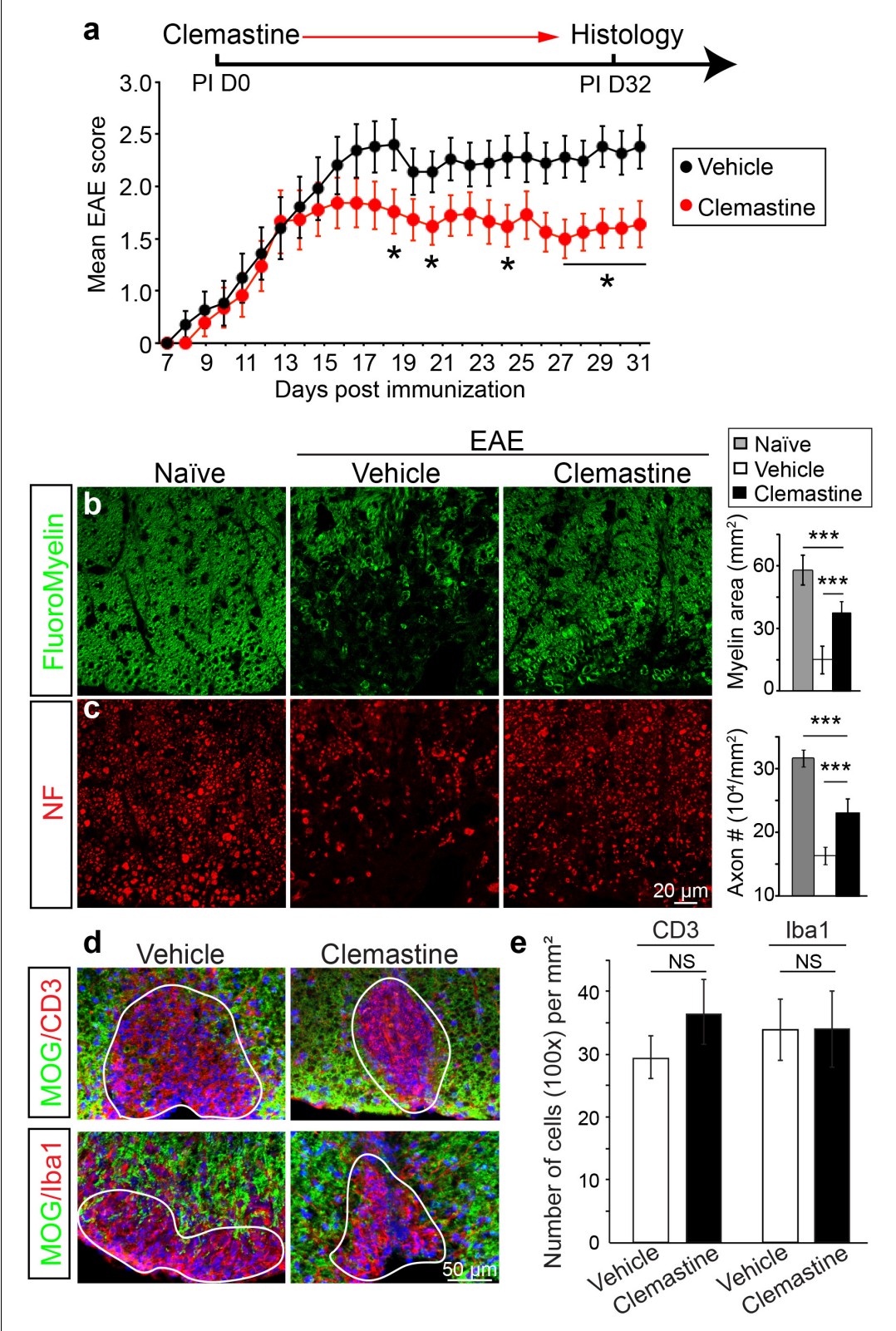

**Figure 2.** Administration of clemastine attenuates EAE clinical scores and prevents axonal loss. (**a**) Daily treatment of clemastine (10 mg/kg, n = 25) from day 0 decreases the clinical severity in the MOG$_{35-55}$ induced EAE model. Error bars represent mean ± s.e.m. Significance of data is based on the Mann-Whitney test for individual days comparing vehicle (n = 25) to clemastine treated (*p<0.05). (**b, c**) FluoroMyelin (green, **b**) and NF immunostaining (red, **c**) illustrates that clemastine promotes myelin positive staining and prevents axonal loss in the white matter tracts of the spinal cord cross sections

*Figure 2 continued on next page*

*Figure 2 continued*

compared to the vehicle control. FluoroMyelin positive areas and NF positive axons were quantified. Error bars represent mean ± s.e.m. and all experiments were performed in triplicate. ***p<0.001, significance based on Student's t-test with the respective controls. n = 3 for all experiments. (**d**) Demyelinated lesions from spinal cord sections from clemastine treated and vehicle control mice were analyzed for T-cells and macrophages/microglia. Sections were immunostained for MOG (green), CD3 (T-cells; red), and Iba1 (macrophages and microglia; red). (**e**) Density of CD3 and Iba1 positive cells were quantified in demyelinated lesions from vehicle control and clemastine treated mice at the late stage of EAE. Error bars represent mean ± s. e.m. (experiments were performed in triplicate).

Consequently, while we can employ this approach to identify newly generated oligodendrocytes and remyelination in EAE, it does not necessarily allow us to identify all remyelinated axons. Despite the coincidence of remyelination and attenuation of clinical scores upon remyelination, it remains unclear whether preservation of axonal integrity is solely due to remyelination, as none of the small molecules are cell-type specific, and off-target effects may contribute to the attenuation of clinical scores in EAE. Clemastine may modulate the immune system via antagonism of a number of receptors— including the histamine receptor 1 and the muscarinic receptors—expressed by immune cells (*Jutel et al., 2001*; *Johansen et al., 2011*). Therefore, uncoupling remyelination from effects on the immunological response and degenerative process requires cell-specific manipulation of OPCs without affecting other cell types. As clemastine is known to possess anti-muscarinic properties and was identified in a cluster of other anti-muscarinic compounds from a high-throughput screen (*Mei et al., 2014*); identification and knockout of the specific target receptor in OPCs could phenocopy the effects elicited by administration of clemastine and allow us to definitively demonstrate the neuroprotective role of remyelination in EAE.

## Identification of the muscarinic receptor target for remyelination

Identification of the target receptor(s) of clemastine may provide an ideal strategy for cell-specific manipulation in OPCs. Clemastine and benzatropine are two potent myelin regenerative compounds that act to antagonize six overlapping targets, including the five muscarinic acetylcholine receptor subtypes (*Chrm1-Chrm5*) and the histamine receptor 1 (*Hrh1*) (*Cohen and Almazan, 1994*; *Cohen et al., 1996*; *De Angelis et al., 2012*; *Kubo et al., 1987*; *Ragheb et al., 2001*). The expression of these receptors on oligodendroglia was examined by analyzing mRNA and antibody staining. qRT-PCR and immunostaining detected *Chrm1-Chrm5* and *Hrh1* expression in OPCs at both mRNA (*Figure 4—figure supplement 1a,b*) and protein levels (*Figure 4—figure supplement 1c,d*). As all the potential receptor targets were expressed by OPCs, albeit at varying levels, we hypothesized that the effects of muscarinic receptor (MR) antagonists on differentiation should be blocked upon loss-of-function. To investigate the potential target(s), OPCs were systematically purified from Chrm1-Chrm5 or Hrh1 knockout mice. After treatment with clemastine, benzatropine, or vehicle, the numbers of MBP-positive OLs and PDGFRα-positive OPCs were quantified (*Figure 4a*). Treated groups were normalized to vehicle controls to reveal the effects of MR antagonists on oligodendroglial differentiation upon individual receptor deletion. As expected, clemastine or benzatropine induced an approximate five-fold increase in the number of OLs and a simultaneous decrease in the number of OPCs when examined in wildtype OPCs (*Figure 4a*). Similar effects of MR antagonists were observed in the cultures from Chrm2-Chrm5 or Hrh1 knockout OPCs, suggesting that none of these individual receptors are essential for mediating the effects of the MR antagonists (*Figure 4a*). Interestingly, knockout of the Chrm1 completely abolished the effects of the anti-muscarinic compounds, resulting in similar numbers of OPCs and differentiated OLs as compared to vehicle-treated Chrm1KO cells, suggesting that the Chrm1 may be the sole mediator of the effects of the MR antagonists on oligodendroglia (*Figure 4a*). To determine if Chrm1 is sufficient for mediating the effects of anti-muscarinic drugs, we examined whether Chrm1 deletion would phenocopy the enhancement of OPC differentiation and myelination by muscarinic antagonists. OPCs from the Chrm2- Chrm5 or Hrh1 knockout mice revealed similar levels of differentiation (MBP-positive) and proliferation (PDGFRα-positive) as compared to the wildtype OPCs (*Figure 4b,c*). However, the Chrm1 knockout cultures displayed a five-fold increase in MBP-positive OLs as well as a significant decrease in the number of OPCs when compared to wildtype cultures, consistent with the hypothesis that deletion of the Chrm1 is sufficient to enhance differentiation of OPCs (*Figure 4c*). To test whether OPCs

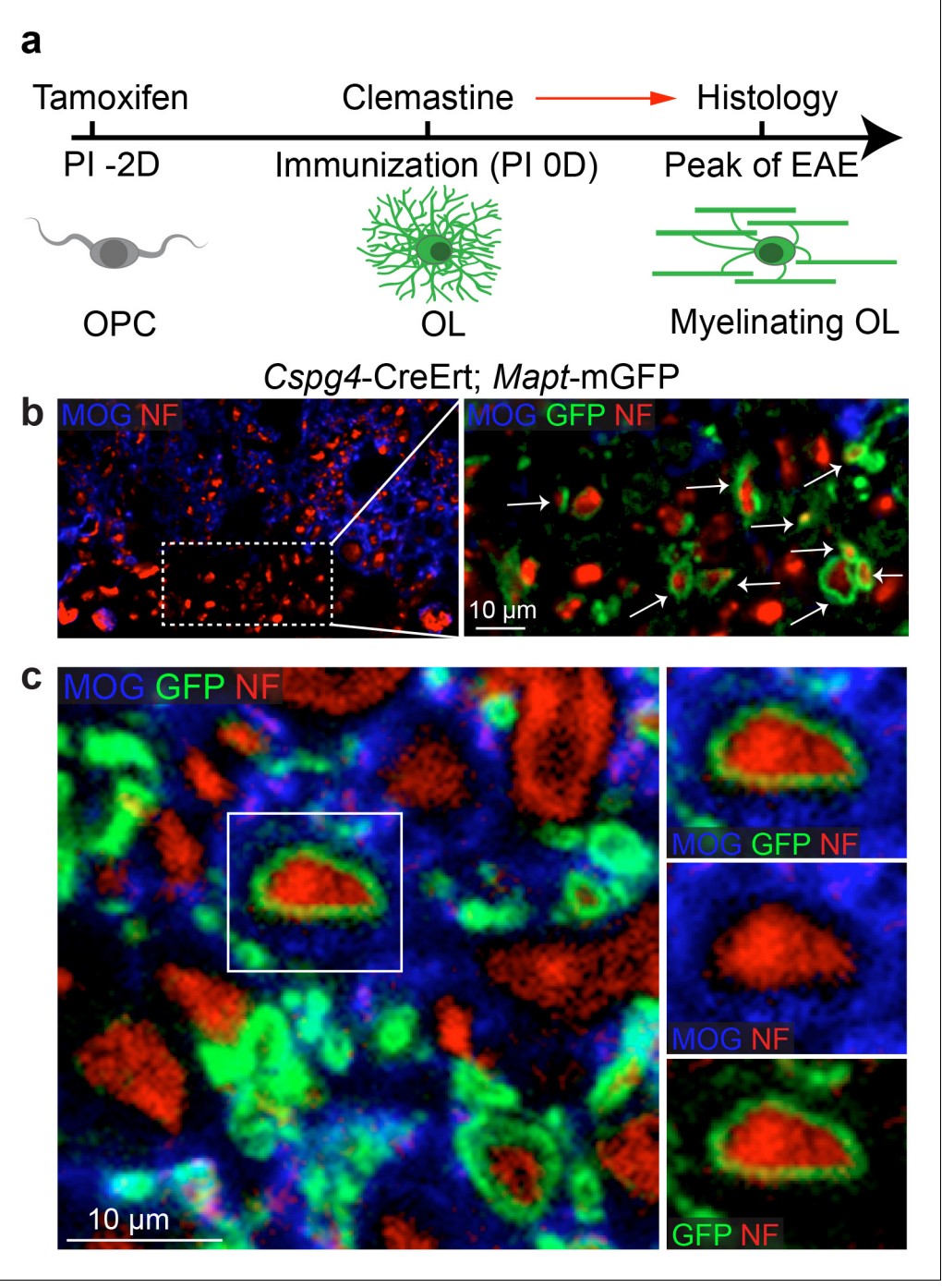

**Figure 3.** Lineage-specific tracing of remyelination by newly-formed oligodendrocytes in EAE. Lineage-specific tracing of remyelination by newly-formed oligodendrocytes was accomplished using the *Cspg4*-CreErt; *Mapt*-mGFP line. Recombination is induced by a single dose of tamoxifen in NG2 cells (OPCs), two days prior to immunization and clemastine administration (**a**). As OPCs do not express tau, only the newly differentiated OLs express mGFP. (**b**) MOG and NF immunostaining display a number of GFP positive myelin sheaths wrapping demyelinated axons (NF+/MOG-) after clemastine treatment. Additionally, a number of axons undergoing remyelination are surrounded by thin GFP membrane that are still yet negative for MOG expression (**c**).

The following figure supplement is available for figure 3:

**Figure supplement 1.** Oligodendroglial lineage-specific tracing of remyelination in EAE treated with vehicle.

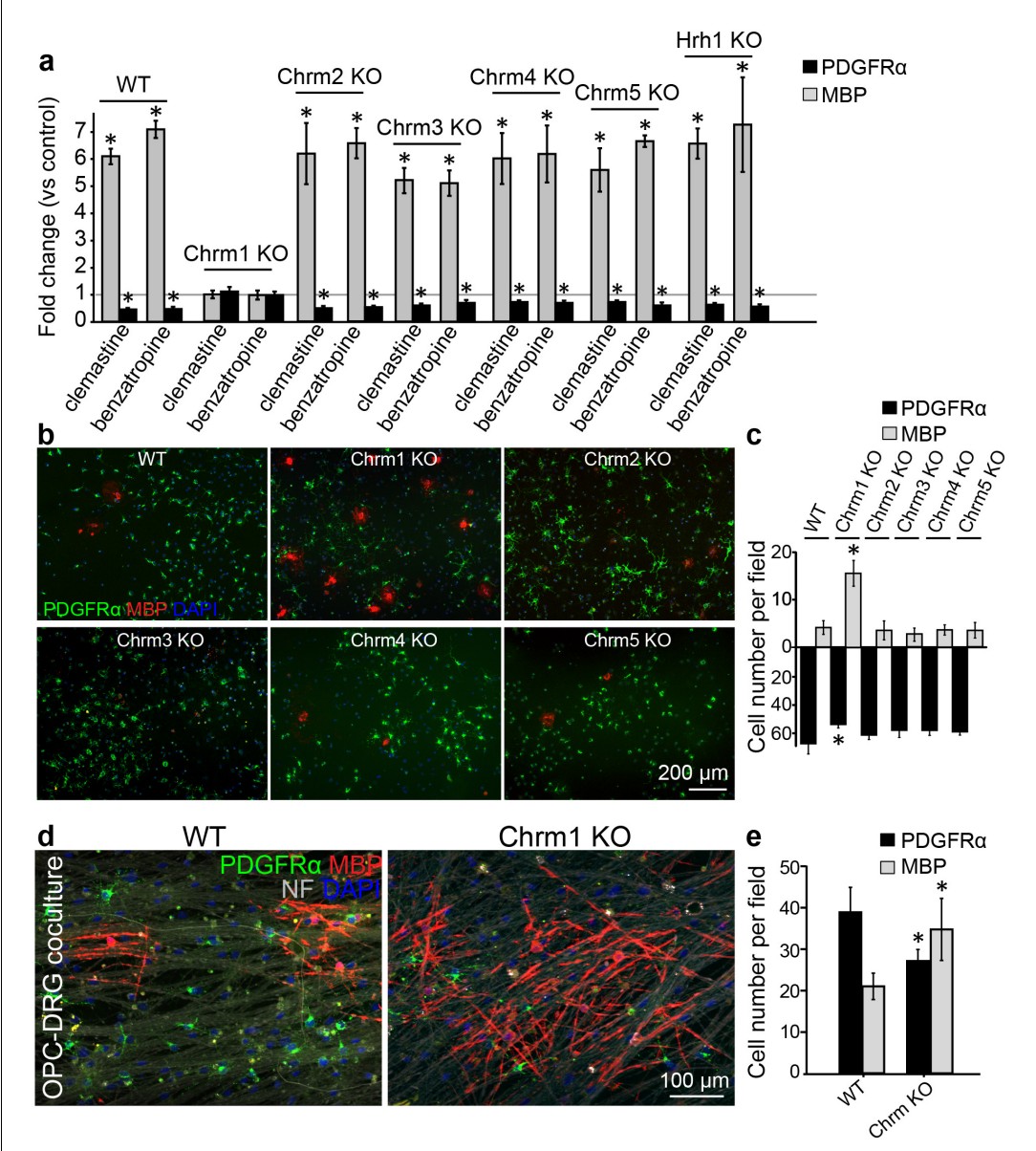

**Figure 4.** Identification of the M1 muscarinic acetylcholine receptor as a target for remyelination. (**a,b**) OPC cultures from Chrm1, Chrm2, Chrm3, Chrm4, Chrm5 or Hrh1 knockout mice are immunostained for MBP (red) and PDGFRα (green). MBP and PDGFRα positive cells are quantified and normalized to the vehicle control in each OPC culture upon 48 hr treatment with clemastine or benzatropine (**a**). Representative images of individual knockout OPC cultures without any treatment (**b**). (**d**) Rat DRG neuron cocultures with Chrm1 null OPCs were cultured for 9 days and immunostained for MBP (red), PDGFRα for OPCs (green) and neurofilament for axons (white). Cell nuclei are identified by DAPI (blue). Quantification of the percentages of MBP- and PDGFRα- positive cells from the purified OPC cultures (**c**) or DRG cocultures (**e**). Error bars represent mean ± s.e.m. and all experiments were performed in triplicate. *$p < 0.05$, significance based on Student's t-test with the respective controls. n = 3 for all experiments.

The following figure supplement is available for figure 4:

**Figure supplement 1.** Expression of muscarinic acetylcholine receptors on oligodendroglia.

from the Chrm1 knockout mice display precocious differentiation and myelination, we cocultured Chrm1 null OPCs with rat dorsal root ganglion (DRG) neurons. We detected a significant increase in number of myelinating OLs as well as a concomitant decrease in OPCs (*Figure 4d,e*). Together, our

results indicate that Chrm1 is a potent negative regulator of differentiation and myelination by oligodendroglia in vitro.

## Deletion of the Chrm1 as an approach to accelerate the kinetics of remyelination

Given the enhanced intrinsic abilities of Chrm1 null OPCs to differentiate and myelinate (*Figure 4*), we hypothesized that the kinetics of remyelination would be accelerated in Chrm1 knockout mice. To test this hypothesis, we implemented the lysolecithin-induced focal demyelination model. Demyelination was induced in the dorsal funiculus and ventrolateral white matter tracts of spinal cords of 6-week old littermates (*Figure 5a*). Remyelination in this model is a process that involves OPC recruitment and differentiation. Based on our previous findings, OPC differentiation and remyelination occur between 7–14 days post lesion (dpl) (*Fancy et al., 2011*; *Mei et al., 2014*). Therefore, we analyzed oligodendrocyte differentiation at 10 dpl (*Figure 5b–e*) by in situ hybridization and immunostaining. *Plp* in situ hybridization illustrated a significant increase in *Plp*-positive OLs in the lesion of Chrm1 null mice when compared to wildtype littermates (*Figure 5b,d*). In addition, both *Mag* in situ hybridization (*Figure 5c*) and MBP immunostaining (*Figure 5e*) of the lesions revealed enhanced differentiation of OLs at 10 dpl in Chrm1 knockout mice. These findings indicate that Chrm1 null OPCs exhibit enhanced differentiation and accelerated kinetics of remyelination after demyelination.

To determine whether Chrm1 is a potential therapeutic target after inflammatory demyelination, we induced EAE in sex matched Chrm1 knockout and wildtype littermates by immunization with MOG$_{35-55}$. Wildtype MOG$_{35-55}$ EAE mice developed chronic paralysis within 8–14 days without an obvious remission period (*Figure 5f*). In contrast, Chrm1 null EAE mice displayed a moderately reduced disease severity from the earliest stages of disease, followed by a remission phase with a significant decrease in clinical score from 19 days after immunization (*Figure 5f*). To test whether Chrm1 deletion promotes remyelination and preserves axonal integrity in EAE mice, myelin was assessed by MOG immunostaining and axons were evaluated by NF immunostaining in the lumbar spinal cord (*Figure 5g,h*). Chrm1 knockout mice exhibited reduced areas of demyelination (indicated by MOG immunoreactivity), suggestive of remyelination and preservation of axons when compared to wildtype EAE mice (*Figure 5i,j*). To examine the possibility that Chrm1 deletion in cells other than oligodendroglia contributes to the prevention of axon loss, we assessed unmyelinated axons in the spinal cord, which were exposed to a similar inflammatory milieu as the myelinated axons. CGRP positive unmyelinated axons in the dorsal horn of the spinal cord were similar in density between the naïve wildtype and Chrm1 knockout mice (*Figure 5k,l*). When subjected to EAE, the density of CGRP positive fibers was dramatically decreased in both Chrm1 null and wildtype mice (*Figure 5k,l*). It is noteworthy that the decrease in the CGRP positive fibers is similar between wildtype and Chrm1 knockout mice, indicating that Chrm1 deletion does not rescue unmyelinated axons from degeneration and is suggestive of the possibility that accelerated remyelination may protect axons from degeneration after EAE. While these results indicate that the Chrm1 represents a potentially effective therapeutic target for EAE, we cannot completely exclude the possibility that the global knockout of Chrm1 diminishes inflammation or is directly axon-protective in some other manner. Therefore, it was imperative to generate the conditional deletion of *Chrm1* specifically in OPCs, as this would permit us to genetically regulate remyelination in EAE and uncouple the immunological response and degenerative process from the impact of oligodendrocyte-mediated repair.

## Accelerated remyelination prevents axonal loss in EAE

To determine whether remyelination is sufficient to prevent axonal loss after inflammatory demyelination, we generated conditional Chrm1 knockout mice specifically in OPCs by crossing the *Chrm1* floxed line with the *Cnp*-Cre line. We initially examined the development of oligodendroglia and myelination in 8-week old mice to determine if the Chrm1 cKO mice would display abnormal myelin. The Chrm1 cKO (*Cnp*-Cre; *Chrm1* fl/fl) mice displayed similar morphologies of axons and myelin as compared to control (*Chrm1* fl/fl) mice by EM. We measured the *g*-ratios of myelinated axons in Chrm1 cKOs and littermate controls (*Figure 6a*). No significant difference was found in *g*-ratios between Chrm1 cKOs and controls (*Figure 6a*). MOG and NF immunostaining showed comparable densities of myelinated fibers in the Chrm1 cKO and control spinal cords (*Figure 6b*). Additionally, the density of differentiated oligodendrocytes was not significantly altered between Chrm1 cKOs

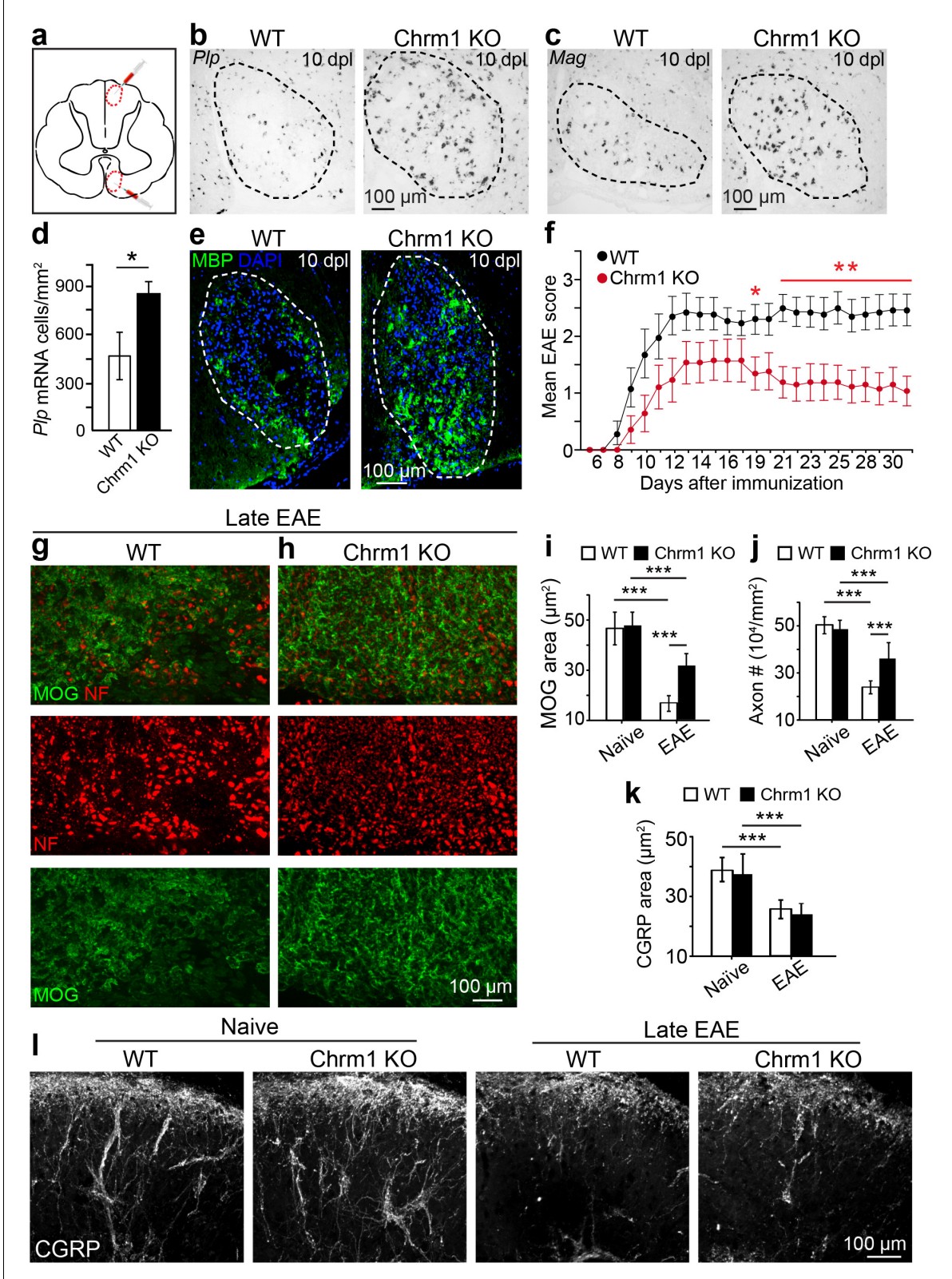

**Figure 5.** Chrm1 deletion accelerates remyelination and attenuates EAE clinical scores. (a) Schematic diagram displays the focal demyelinated lesions in the dorsal funiculus and ventrolateral white matter of mouse spinal cord. (b–e) Mice were analyzed by in situ hybridization of *Plp* (b) or *Mag* (c) and by MBP staining (e) at 10 days post lesion (dpl). (d) Quantification of *Plp* in situ hybridization displays a two-fold increase in *Plp* positive OLs. Error bars represent mean ± s.d. (4 animals in each group), and all experiments were performed in quadruplicate. (f) The clinical severity in MOG$_{35-55}$ induced EAE
*Figure 5 continued on next page*

*Figure 5 continued*

was reduced in Chrm1 null mice (n = 14) as compared to sex matched wildtype (n = 15) littermates. Cumulative data shown represent mean ± s.e.m. from two independent experiments. Significance based on Mann-Whitney test for individual days, *p<0.05 or **p<0.01. (g, h) Spinal cord sections immunostained for MOG (green) and NF (red) showing myelin and axonal staining in wildtype and Chrm1 null mice. (i, j) Quantification of MOG+ areas (i) and NF+ axons (j) from Chrm1 knockout and wildtype mice with or without EAE induction. (k) Quantification of CGRP+ areas in the dorsal spinal cord from Chrm1 knockout and wildtype mice. (l) Staining and quantification of CGRP+ fibers in Chrm1 knockout and wildtype mice before and after EAE. Error bars (k) represent mean ± s.e.m. (from 3 representative animals in each group) and all experiments were performed in triplicate. *p<0.05; ***p<0.001, significance based on Student's t-test with the respective controls.

and controls (*Figure 6c*). These results suggest normal development of nerve fibers in adulthood of Chrm1 cKO mice (*Figure 6*). We next induced EAE in sex matched Chrm1 cKO, Chrm1 heterozygous

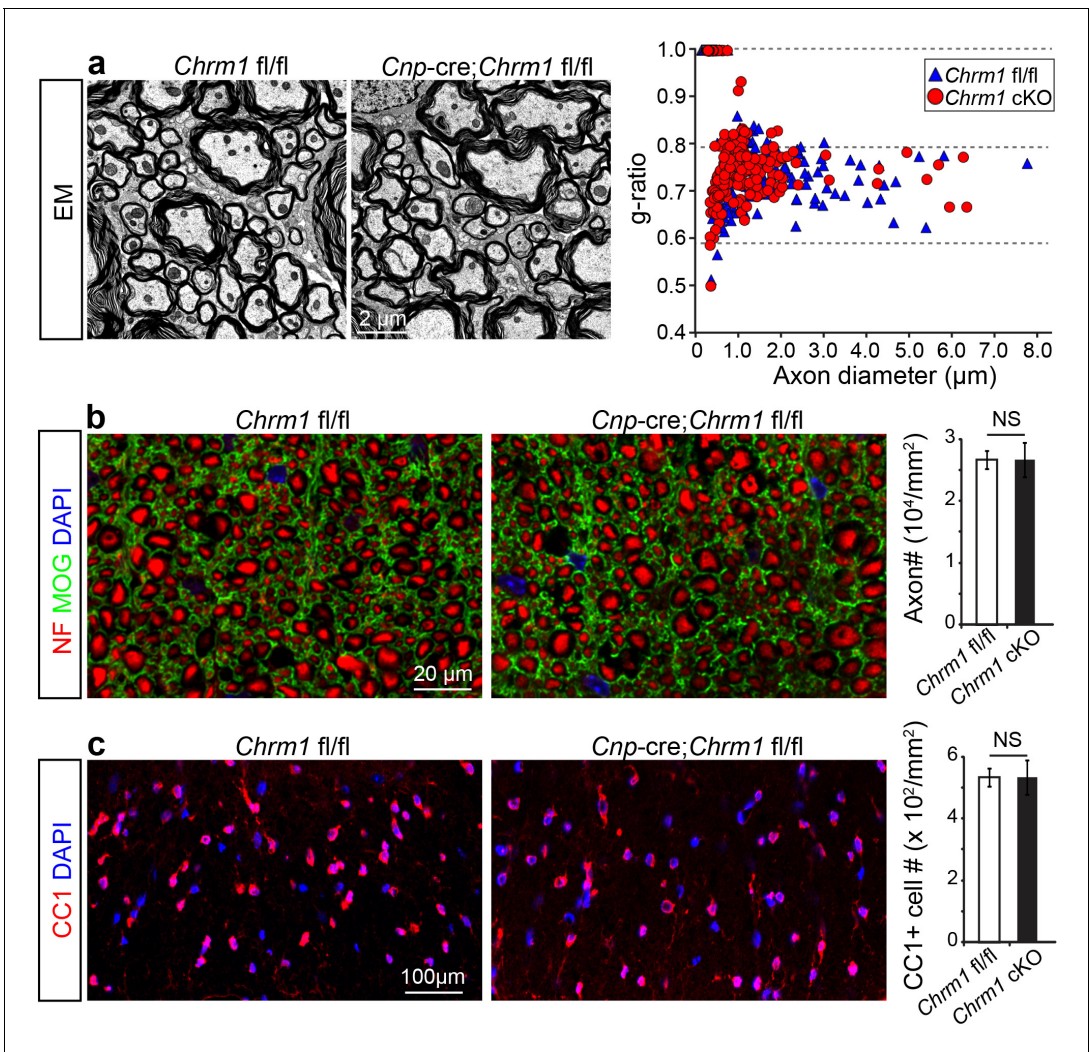

**Figure 6.** Myelin ultrastructure in Chrm1 cKO spinal cords is normal in adult mice. (a) Representative electron microscopic images of spinal cord white matter tracts from control (*Chrm1* fl/fl) and Chrm1 cKO mice at 8 weeks of age. The scatterplot displays *g*-ratios of individual axons as a function of axonal diameter. All *g*-ratios were analyzed from transmission electron microscopic images. (b) Analysis of MOG and NF immunostaining of the spinal cord white matter tracts from control and Chrm1 cKO mice. Myelinated axons in the spinal cord white matter tracts were quantified. (c) Immunostaining for CC1 displays mature oligodendrocytes in the spinal cord white matter tracts of control and Chrm1 cKO mice. Quantification of CC1 positive cell numbers in the spinal cord white matter tracts. Error bars represent mean ± s.e.m. (3 representative mice from each group), and all experiments were performed in triplicate. Significance based on Student's t-test with the respective controls.

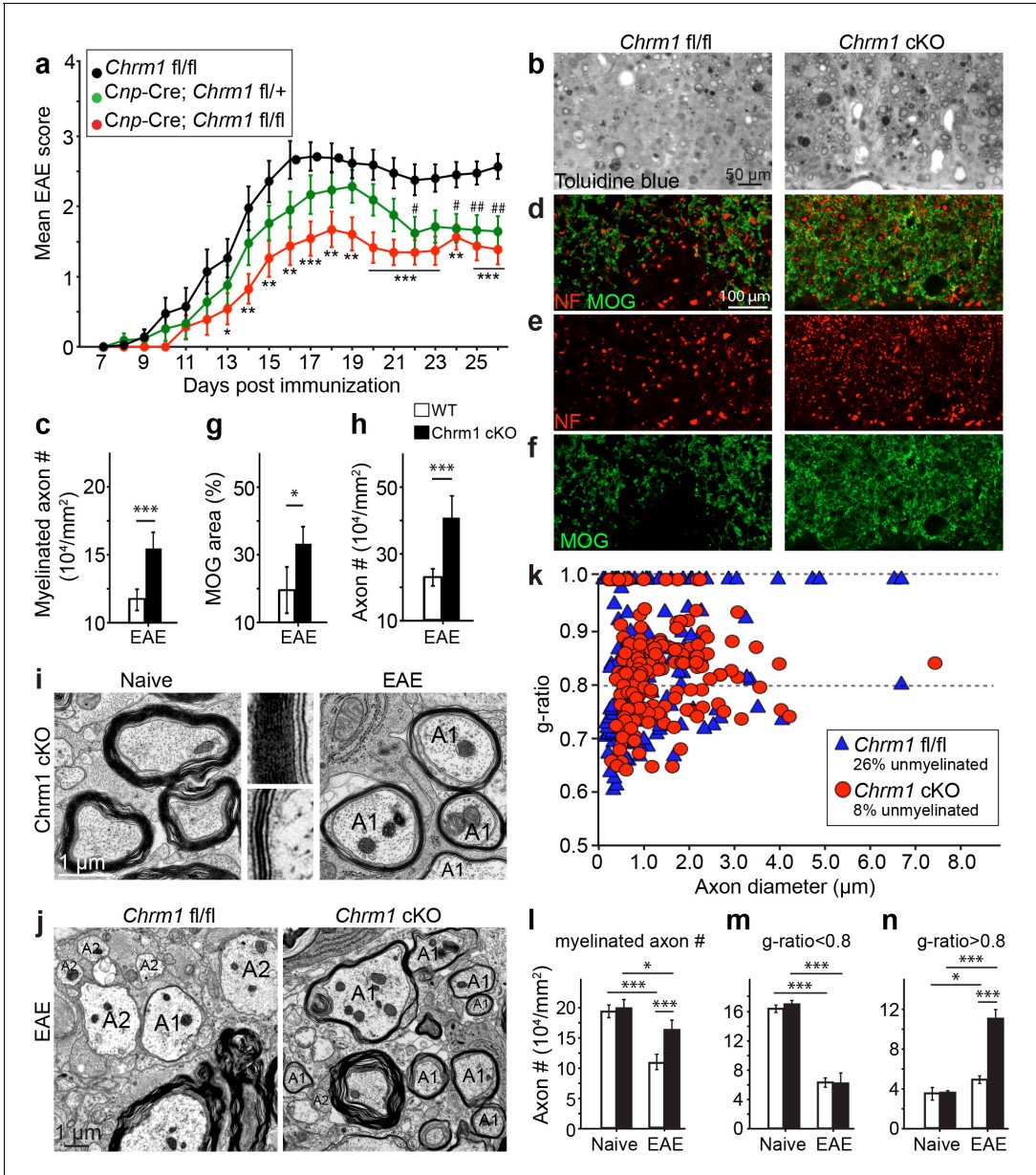

**Figure 7.** Accelerated remyelination prevents axonal loss and restores neuronal function in EAE. (a) The clinical severity in MOG$_{35-55}$ induced EAE was reduced in the Chrm1 conditional knockout mice (*Cnp*-cre; *Chrm1* fl/fl, n = 27) and Chrm1 conditional knockout heterozygotes (*Cnp*-cre; *Chrm1* fl/+, n = 24) as compared to control (*Chrm1* fl/fl, n = 23) littermates. Cumulative data shown represent mean ± s.e.m. from four independent experiments. Significance based on the Mann-Whitney test for individual days, *p *or* # < 0.05, **p or ## < 0.01 or ***p< 0.001. (b) Spinal cord semi-thin sections stained with toluidine blue from Chrm1 cKO and control EAE mice. (d–f) Spinal cord cross sections stained with MOG (green) and NF (red) showing myelin and axonal density in Chrm1 cKO and control EAE mice (d–f). (c, g, h) Quantification of myelinated axons (c), MOG+ density (g) and NF+ axon numbers (h) from the Chrm1 cKO and control EAE mice. (i) Representative electron microscopic image of pre-existing myelinated axons and remyelinated axons (A1) in the spinal cord white matter tracts of Chrm1 cKO naïve and EAE mice respectively. (j) Remyelinated (A1) and demyelinated (A2) axons in the EAE Chrm1 cKO and control mice. (k) Quantification of myelin sheath thickness and the proportion of myelinated and unmyelinated axons in Chrm1 cKO (red) and control (blue) EAE mice. The scatterplot displays *g*-ratios of individual axons as a function of axonal diameter. Quantification of myelinated axons (i), and axons with *g*-ratio < 0.8 (m) or > 0.8 (n) in the EAE and naïve spinal cord of Chrm1 cKO and control mice. Error bars represent mean ± s.e.m. (3 representative animals in each group), and all experiments were performed in triplicate. Significance based on Student's t-test with the respective controls.

The following figure supplement is available for figure 7:

**Figure supplement 1.** Chrm1 deletion from oligodendroglia does not influence T-cell/macrophage infiltration or microglial activation.

(*Cnp*-Cre; *Chrm1* fl/+) and control littermates by MOG$_{35-55}$ immunization (*Figure 7a*). Control EAE mice developed chronic paralysis within 10–15 days without an obvious remission period (*Figure 7a*). In contrast, disease severity of the Chrm1 cKO mice was significantly attenuated at early stages (PI 13) and throughout the chronic stage (*Figure 7a*). Notably, Chrm1 heterozygous mice revealed a moderately reduced disease severity in the early stages, followed by a remission phase and significant decrease in clinical score from 22 days after immunization (*Figure 7a*). These results demonstrate that the cell-specific deletion of Chrm1 in OPCs is sufficient to attenuate the functional deficits elicited by EAE induction, and that these effects by Chrm1 inactivation may be dose-dependent as the heterozygote and homozygote mice resulted in similar clinical scores at the late stage of EAE.

To investigate whether these effects are due to remyelination and the subsequent preservation of axonal integrity, we examined myelinated axons in semi-thin sections of EAE spinal cords counterstained with toluidine blue (*Figure 7b,c*). Myelinated axon density in the ventral white matter was significantly higher in the Chrm1 cKO EAE spinal cords, as compared to controls (*Figure 7c*). Consistently, immunostaining for MOG and NF showed a significant increase in MOG immunoreactive areas and NF positive axons in the Chrm1 cKO EAE spinal cords, as compared to the controls (*Figure 7d–h*). To determine whether remyelination is neuroprotective and preserves axonal integrity in Chrm1 cKO mice, we examined the axons in the spinal cord ventral white matter with EM (*Figure 7i,j*). Remyelinated axons in the EAE spinal cord were identified by thinner myelin sheaths, as compared to the pre-existing myelinated axons with thicker myelin sheaths in naïve mice (*Figure 7i*). In the control EAE mice, demyelinated and degenerating axons were easily detected with a number of pre-existing myelinated axons and just a few remyelinated axons. In contrast, remyelinated axons were more consistently observed in the Chrm1 cKO EAE spinal cords (*Figure 7j*). We next quantified the *g*-ratio of the axons in the EAE spinal cords, focusing on the ventral white matter (*Figure 7k*). The percentage of unmyelinated axons (*g*-ratio = 1) was decreased in the Chrm1 cKO EAE spinal cords, suggesting that remyelination is enhanced in the Chrm1 cKO EAE mice (*Figure 7k*). The overall myelinated axon density (per mm$^2$) was significantly decreased in the EAE control mice as compared to the Chrm1 cKO mice, suggesting that less axons undergo degeneration in Chrm1 cKO EAE mice (*Figure 7l*). To determine whether remyelination is the direct cause of enhanced axonal survival, we sorted remyelinated axons from pre-existing myelinated axons by examining *g*-ratios. We reasoned that axons with a *g*-ratio greater than 0.8 were likely to be remyelinated axons (*Figure 7k*), since most of the axons in the naïve spinal cord white matter exhibited *g*-ratios below 0.8 (pre-existing myelinated axons) (*Figure 6a*) (*Mei et al., 2014*). The pre-existing myelinated axon density (*g*-ratio < 0.8) significantly decreased in the EAE mice, without any significant difference between the Chrm1 cKO and control EAE mice, suggesting that Chrm1 deletion in OPCs does not affect the severity of demyelination (*Figure 7m*). Remyelinated axon density (*g*-ratio >0.8) was detected in control EAE mice and was significantly increased as compared to naïve mice, suggesting that remyelination is an ongoing process even in the control EAE mice (*Figure 7n*). Notably, a two-fold increase in the remyelinated axon density was detected in the Chrm1 cKO mice as compared to control EAE mice (*Figure 7n*), demonstrating that remyelination is indeed neuroprotective from the neuroinflammatory environment in EAE. Additionally, we examined the infiltration of T-cells, macrophages and the activation of microglia by immunostaining for CD3 and Iba1 and quantified the number of these cells in demyelinated lesions of the Chrm1 cKOs and control littermates at the late stage of EAE (*Figure 7—figure supplement 1*). We did not find a statistical difference in the number of CD3 or Iba1 positive cells in lesions from the Chrm1 cKOs and the control littermates, suggesting that the Chrm1 cKO does not overtly alter the inflammatory response (*Figure 7—figure supplement 1*). While we cannot exclude the possibility that deletion of Chrm1 from OPCs may alter a yet unidentified inflammatory role by OPCs, taken together, our findings suggest that the most likely scenario is that remyelination is playing a causal role in preserving axon integrity after inflammatory demyelination.

## Discussion

Axonal degeneration underlies the chronic disability and progression in MS, and is currently untreatable (*Bitsch et al., 2000*; *Bjartmar et al., 2003*; *Franklin et al., 2012*; *Kuhlmann et al., 2002*; *Lovas et al., 2000*; *Tallantyre et al., 2009*; *Trapp and Stys, 2009*). Immunomodulatory approaches

are effective in decreasing the number of relapses and may impact the timing of progressive disability but fail to fully prevent progression (*Carrithers, 2014*; *Rice, 2014*). Currently, no intervention is available for neuroprotection, especially as the the mechanisms resulting in axonal degeneration are not fully understood (*Bjartmar and Trapp, 2003*; *Sherman and Brophy, 2005*; *Wang et al., 2012*). How then can we prevent axon degeneration after inflammatory demyelination? Emerging evidence indicates that the myelin sheath is important to ensure the survival of axons by providing physical and metabolic support in the developing CNS (*Hirrlinger and Nave, 2014*; *Morrison et al., 2013*; *Saab et al., 2013*). Since remyelination is an inefficient process in MS lesions, remyelinating therapies have been proposed as a potential therapeutic approach for MS (*Franklin and Ffrench-Constant, 2008*). In order to promote myelin repair, recent high-throughput screening efforts have yielded a number of myelin regenerative compounds (*Deshmukh et al., 2013*; *Mei et al., 2014*; *Najm et al., 2015*), and interestingly some of these compounds are capable of attenuating the clinical severity of EAE (*Deshmukh et al., 2013*; *Najm et al., 2015*). However, our limited understanding of the significance of remyelination dampens enthusiasm for myelin regenerative approaches to protect axons and prevent progression, as attenuation of EAE clinical symptoms may also be attributed to immunomodulatory effects from nonselective pharmacotherapeutic approaches. Here we utilize a genetic approach to enhance remyelination by manipulating oligodendroglia specifically without affecting other cell types, namely by the conditional deletion of Chrm1 in OPCs. For the first time, our findings uncouple remyelination from active inflammation and axonal degeneration in EAE and clearly demonstrate that remyelination is sufficient to preserve axonal integrity and neuronal function under an inflammatory demyelinating condition. It is notable that endogenous remyelination is nearly absent at the peak of EAE and is only sparsely observed at the later stages, suggesting cell autonomous remyelination is an inefficient process in the inflammatory setting. Strikingly, myelin regenerative approaches are able to initiate remyelination as early as the peak of EAE, as demonstrated by lineage tracing, and is sufficient to rescue axons from degeneration, possibly by reconstructing metabolic and/or physical support for demyelinated axons. Additionally, we hypothesize that remyelinated axons are excluded from the MOG-induced demyelination due in part to the absence of MOG expression in the immature newly formed myelin. This is clearly evident upon the detection of a number of axons surrounded by thin GFP membranes that are still yet negative for MOG expression (*Figure 3c*). Furthermore despite the thin myelin observed after remyelination, this process appears to be sufficient to preserve a large number of axons. This evidence suggests that early remyelination could be an effective intervention to preserve axonal integrity after inflammatory demyelination. Together, our findings provide novel insights into the functional role of remyelination as a means to preserve axonal integrity and neuronal function after inflammatory demyelination. Our findings clearly identify Chrm1 on OPCs as the target of anti-muscarinic treatment for remyelination – suggesting that remyelination therapy is a promising approach for restoring function and prolonging the quality of life for patients with MS.

## Material and methods

### Transgenic mice

Chrm1, Chrm2, Chrm3, Chrm4, Chrm5, and Hrh1 knockout mice have been described previously (*Noubade et al., 2007*; *Wess, 2004*). The Chrm1-Chrm5 knockout mice were backcrossed for at least 10 generations onto the C57BL/6 genetic background. In Chrm1 mutant mice, Chrm1 function was abolished by replacing a genomic fragment that included the translation start site and the region coding for the first 54 amino acids of the Chrm1 protein with a PGK-neomycin resistance cassette (*Gerber et al., 2001*). *Chrm1* +/- mice were crossed to generate Chrm1 null (*Chrm1* -/-) and wildtype (*Chrm1* +/+) mice. *Chrm1* floxed mice were provided from Dr. Susumu Tonegawa (*Kamsler et al., 2010*) and *Cnp*-Cre mice were provided from Dr. Klaus Nave. Both lines have been maintained in C57BL/6 genetic background for more than 10 generations. The breeding strategy for Chrm1 cKO was to cross *Cnp*-Cre; *Chrm1* fl/+ with *Chrm1* fl/fl to generate conditional knockout homozygous (*Cnp*-Cre; *Chrm1* fl/fl), heterozygous (*Cnp*-Cre; *Chrm1* fl/fl) and littermate controls (*Chrm1* fl/fl). The *Cspg4*-CreERT2 mouse (*Zhu et al., 2011*) was kindly provided by Dr. Anders Persson (UCSF, San Francisco) and the *Mapt*-GFP reporter (*Hippenmeyer et al., 2005*) mouse by Dr. John Rubenstein (UCSF, San Francisco). Genotypes of all mice were determined by PCR analysis of

tail genomic DNA using appropriate primers. All mice examined in this study were handled in accordance with the approval of the University of California San Francisco Administrative Panel on Laboratory Animal Care.

## Tamoxifen administration

Tamoxifen (Sigma-Aldrich, St. Louis, MO) was dissolved at 0.5 mg/ml in sunflower oil and one dose (40 µl) was administrated to mice by oral gavage 2 days before immunization with the MOG peptide.

## Purification of oligodendrocyte precursor cells

Immunopanning purification of OPCs was performed as previously described (Lee et al., 2013). Briefly, OPCs were purified from 7–9 postnatal rat or mouse brain cortices. Tissue culture dishes were incubated overnight with goat IgG and IgM secondary antibodies to mouse (Jackson Laboratories, Cat: 115-005-004) in 50 mM Tris-HCL a final concentration of 10 µg ml$^{-1}$, pH 9.5. Dishes were rinsed and incubated at room temperature with primary antibodies for Ran-2, GalC and O4. Rodent brain hemispheres were diced and dissociated with papain (Worthington) at 37°C. After trituration, cells were resuspended in a panning buffer (0.2% BSA in DPBS) and incubated at room temperature sequentially on three immunopanning dishes: Ran-2 and GalC were used for negative selection before positive selection with O4. OPCs were released from the final panning dish using 0.05% Trypsin (Fisher Scientific, Pittsburgh, PA). OPCs are typically 95% pure after immunopanning, with a viability of 94%.

## Immunofluorescence

Mice were deeply anesthetized with 1% pentobarbital and transcardially perfused with 4% paraformaldehyde in PBS. Brains and spinal cords were dehydrated in 30% sucrose and sectioned (10 or 20 µm) on a cryostat microtome (HM450, Thermo). OPCs cultured on coverslips were fixed with 4% paraformaldehyde in PBS. Free floating or coverslips sections were blocked with 20% normal goat serum and incubated with primary antibodies overnight at 4°C. Sections were incubated with secondary antibodies for 1 hr at room temperature. Primary antibodies included the following: rat monoclonal antibody to MBP (1:500, Millipore, Cat: MAB395), rabbit monoclonal antibody to PDGFRα (1:8000, gift from W.B. Stallcup), goat anti-Chrm1 (1:100; Santa Cruz Biotechnology, Cat: sc-7470), goat anti-Chrm3 (1:100; Santa Cruz Biotechnology, Cat: sc-31487), goat anti-Chrm5 (1:100; Santa Cruz Biotechnology, Cat: sc-7479), mouse monoclonal anti-Chrm2 (1:100, Abcam, Cat: ab90805), mouse monoclonal anti-Chrm4 (1:100, Abcam, Cat: ab77956), Goat anti-Hrh1 (1:100, Sigma, Cat: SAB2501418), mouse anti-CGRP (1:10,000, Abcam, Cat: ab81887)), mouse anti-CC1 (1:500, Calbiochem, Cat: OP80), rabbit anti-NF200 (1:200, Abcam, Cat: Ab1987), human anti-MOG (1:1000) (von Büdingen et al., 2008). Secondary antibodies included the following: AlexaFluor-488-, AlexaFluor-568-, or Cy5-conjugated secondary antibodies to rabbit, mouse, or goat (1:500; Invitrogen). Cell nuclei were identified with DAPI (Vector Labs). For myelin staining, sections were incubated with FluoroMyelin (1:500, Invitrogen, Cat: F34651) in PBS for 20 min at RT.

## RT-PCR

RNA was extracted from rat OPC cultures or rat cortex using Trizol (Invitrogen, Cat: 15596026). Reverse transcription was performed using the RETROscript Reverse Transcription Kit (Life Technologies, Cat: AM1710). PCR was performed as described previously (Mei et al., 2016). Briefly, the DNA polymerase (Invitrogen, Cat: 10342–020) was used to amplify the target gene of cDNA. Triplicate samples were analyzed for each cDNA with test and internal control primers for the house keeping gene, glyceraldehyde-3-phosphate dehydrogenase (GAPDH). Forward and reverse primers used for expression analyses are as follows: Chrm1F (agcagcagctcagagaggtc) and Chrm1R (gggcatcttgatcaccactt); Chrm2F (tcgctccgttatgaatctcc) and Chrm2R (tccacagtcctaacccctac); Chrm3F (gtgccatcttgctagccttc) and Chrm3R (tcacactggcacaagaggag); Chrm4F (gacggtgcctgataaccagt) and Chrm4R (ctcaggtcgatgcttgtgaa); Chrm5F (acagagaagcgaaccaagga) and Chrm5R (ctcagccttttcccagtcag); Hrh1F (gagcttcgggaagacaagtg) and Hrh1R (ttggcaccttccttggtatc); and GAPDHF (aggccggtgctgagtatgtc) and GAPDHR (tgcctgcttcaccaccttct).

## qRT-PCR

qRT-PCR was performed using the CFX96 Touch qRT-PCR System (Bio-Rad). RNA was extracted from purified rat OPC cultures by using Trizol (Invitrogen). cDNA was generated using a first-strand cDNA synthesis kit (Promega). The fluorescent dye SYBR Green (Roche) was included in each assay to measure DNA concentration during each annealing phase. Triplicate samples were analyzed for each cDNA with test and internal control primers for the house keeping gene, beta-actin. Dilution curves were generated by 10-fold serial dilutions of each target cDNA to estimate copy number of the target cDNA. Fluorescence intensity was plotted against cycle number and normalized using beta-actin to account for sample variability. Primers used for expression analyses are as follows: Chrm1F (atcaccacaggcctcctgtc) and Chrm1R (aagattcatgacagaggcgttg); Chrm2F (accctctacactgt-gattggcta) and Chrm2R (ggcccagaggatgaaggaa); Chrm3F (agagctggaagcccagtgc) and Chrm3R (gtagcttggtagagttgaggatgg); Chrm4F (gtggtgagcaatgcctctgtc) and Chrm4R (tgaagcactggttatcagg-cac); Chrm5F (cagtgtccaaagacccttcaac) and Chrm5R (gacatagcacagccagtaaccc); beta-actinF (acgtt-gacatccgtaaagacc) and beta-actinR (catcgtactcctgcttgct).

## Image acquisition and quantification

Fluorescent images from cultured oligodendroglia and cocultures were collected on a Zeiss Axio Imager Z1 fluorescence microscope or a Zeiss LSM-700 confocal microscope with the excitation wavelengths appropriate for Alexa Fluor 488 (488 nm), 596 (568 nm), 647 (628 nm) or DAPI (380 nm). For statistical analysis, at least three representative fields (20x) were randomly acquired from each of the wells. Detection and quantification was performed using the Zen software (Zeiss) and the Image-Pro Plus software 5.0 (Media Cybernetics, Silver Spring, MD, USA).

## Electron microscopy

For electron microscopy, animals were perfused with 1.25% glutaraldehyde, 2% paraformaldehyde in 0.1 M sodium cacodylate pH 7.4 after an initial flush with 0.1 M sodium cacodylate pH 7.4. Tissue was then processed at the Electron Microscopy Core facility, Gladstone Institutes (UCSF). Tissue was post fixed in 2% osmium tetroxide in the same buffer, blocked and stained with 2% aqueous uranyl acetate, dehydrated in acetone, infiltrated and embedded in LX-112 resin (Ladd Research Industries, Burlington, VT). Semi-thin sections were stained with toluidine blue. Samples were next ultrathin sectioned on a Reichert Ultracut S ultramicrotome and counter stained with 0.8% lead citrate. Grids were examined on a JEOL JEM-1230 transmission electron microscope (JEOL USA, Inc., Peabody, MA) and photographed with the Gatan Ultrascan 1000 digital camera (Gatan Inc., Warrendale, PA). G-ratios of myelinated fibers were calculated as the ratio of the diameter of the axon to the diameter of the axon with the myelin sheath, measured using Image-Pro Plus software. Measurements were made on electron micrographs from three pairs of mice in all cases.

## OsO4 staining

Spinal cord sections were post-fixed with 2% glutaraldehyde in 0.1 M PBS, pH 7.2 for 1 hr. Samples were rinsed in PBS and sections were incubated with 0.1% OsO4 in PBS at RT for 45 min.

## Oligodendrocyte precursor cell–dorsal root ganglion cocultures

OPC-DRG cocultures were prepared as previously described (*Mei et al., 2014*, *2016*). Briefly, DRG neurons from E15 Sprague-Dawley rats were dissociated, plated (150,000 cells per 25 mm cover glass) and purified on collagen-coated coverslips in the presence of 100 ng ml-1 NGF (AbD Serotec). Neurons were maintained for 3 weeks and washed with DMEM (Invitrogen) extensively to remove any residual NGF before seeding OPCs. Cocultures were grown in chemically defined medium composed of DMEM (Invitrogen) supplemented with B27 (Invitrogen), N2 (Invitrogen), penicillin-streptomycin (Invitrogen), N-acetyl-cysteine (Sigma-Aldrich) and forskolin (Sigma-Aldrich).

## Lysolecithin-induced demyelination in the spinal cord

Demyelinated lesions were induced in the dorsal funiculus and the ventrolateral white matter regions of the spinal cord of 8-week-old wildtype and Chrm1 null littermates as previously described (*Chong et al., 2012*). Briefly, the animals were anesthetized with isofluorane and buprenorphine and the spinal cords were exposed at level T12/13. 0.5 µL of 1% lysolecithin (l-a-lysophosphatidylcholine)

was administered with a Hamilton needle for each lesion site. Four wildtype mice and four Chrm1 null mice were injected and analyzed at 10 d.p.l.

## EAE induction and clinical score

EAE was induced in female C57BL/6 mice (Jackson Lab, Cat: 000664) at 8–10 weeks of age by subcutaneous flank administration of 100 µg of myelin oligodendrocyte glycoprotein (MOG) peptide (amino acid 35–35; Auspep) in Complete Freund's Adjuvant (CFA) containing 2 mg/ml non-viable Mycobacterium tuberculosis H37 RA (DIFCO). 200 ng pertussis toxin (List Biological Laboratories Inc.) was administered intraperitoneally on day 0 and 2. $MOG_{35-55}$ induced EAE mice were treated daily with 10 mg/kg clemastine (in PBS with 10% DMSO by oral gavage) or the equivalent volume of vehicle. For Chrm1 KO or Chrm1 cKO mice or sex matched wildtype littermate control mice (on a C57BL/6 genetic background), EAE was induced by immunization with 100 µg of myelin oligodendrocyte glycoprotein (MOG) peptide (amino acid 35–55; Auspep) and 200 ng pertussis toxin on day 0 and 2 in both male and female mice at 8–10 weeks of age. The *Cspg4*-CreErt; Tau-GFP mice were immunized with 33 µg of recombinant mouse MOG (rmMOG) protein. Mice were scored daily as follows: 0 = no signs; 0.5 = distal limp tail; 1 = limp tail; 1.5 = inability to turn immediately when flipped on the back; 2 = waddling gait; 2.5 = bilateral hind limb paresis; 3 = severe bilateral hind limb paresis with paralysis of one hind limb; 3.5 = bilateral hind limb paralysis; 4 = beginning fore limb paresis; 4.5 = severe fore limb paresis (animal were euthanized); 5 = moribund.

## Statistical analyses

Statistical analyses were performed on clinical data from EAE experiments using the Mann-Whitney test to examine individual days. Otherwise, two-tailed Student's t-test was used to determine the significance that is expressed as * or #$p<0.05$, ** or ##$p<0.01$ or ***$p<0.001$ compared to control cultures or control mice (vehicle or wildtype). The investigators were blinded to allocation of knockout mice in the EAE experiment until the final statistical analysis.

## Acknowledgements

We thank the MS Research Group at UCSF for their support and the members of the Chan Laboratory for critically reading the manuscript, advice and insightful discussions. This work was supported by the US National Multiple Sclerosis Society (RG5203A4), NIH/NINDS (R01NS062796), Target ALS (A121679) and the Rachleff Endowment and gifts from the friends of the MS Research Group to JRC, the Overseas Youth Program Grant of the Third Military Medical University and the National Natural Science Foundation of China (NSCF31471043, 81270017) to FM, fellowship grants from the Deutsche Forschungsgemeinschaft (DFG; Le 3079/1-1) and the US National Multiple Sclerosis Society (FG 2067-A-1) to KL-H, the Joint Research Fund for Overseas Chinese Young Scholars (NSCF 31228011) and the Chongqing Scientific and Technical Innovation Foundation of China (CSTCKJCXLJRC07) to LX.

## Additional information

### Funding

| Funder | Grant reference number | Author |
| --- | --- | --- |
| Third Military Medical University | Overseas Youth Program Grant | Feng Mei |
| National Natural Science Foundation of China | NSCF31471043 | Feng Mei |
| National Natural Science Foundation of China | 81270017 | Feng Mei |
| Deutsche Forschungsgemeinschaft | Fellowship grants (DFG; Le 3079/1-1) | Klaus Lehmann-Horn |
| National Multiple Sclerosis Society | FG 2067-A-1 | Klaus Lehmann-Horn |

| Joint Research Fund | NSCF 31228011 | Lan Xiao |
| Chongqing Scientific and Technical Innovation Foundation of China | CSTCKJCXLJRC07 | Lan Xiao |
| National Multiple Sclerosis Society | RG5203A4 | Jonah R Chan |
| Target ALS | A121679 | Jonah R Chan |
| National Institutes of Health | R01NS062796 | Jonah R Chan |
| Rachleff Endowment | | Jonah R Chan |

The funders had no role in study design, data collection and interpretation, or the decision to submit the work for publication.

## Author contributions

FM, KL-H, JRC, Conception and design, Acquisition of data, Analysis and interpretation of data, Drafting or revising the article, Contributed unpublished essential data or reagents; Y-AAS, KAR, KP, SAS, SPJF, Acquisition of data, Analysis and interpretation of data; KJS, DSL, Acquisition of data, Contributed unpublished essential data or reagents; LX, H-CvB, Analysis and interpretation of data, Contributed unpublished essential data or reagents; CT, JW, JJL, Contributed unpublished essential data or reagents; AJG, Analysis and interpretation of data, Drafting or revising the article; SSZ, Analysis and interpretation of data, Drafting or revising the article, Contributed unpublished essential data or reagents

## Author ORCIDs

Jonah R Chan, http://orcid.org/0000-0002-2176-1242

## Ethics

Animal experimentation: All mice examined in this study were handled in accordance with the approval of the University of California San Francisco Administrative Panel on Laboratory Animal Care (APPROVAL NUMBER: AN097874-03B). All surgeries were terminal and performed under University guidelines–with every effort to minimize suffering.

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
