## [Decision Letter]

Thank you for submitting your article "Accelerating Remyelination During Inflammatory Demyelination Prevents Axonal Loss and Improves Functional Recovery" for consideration by *eLife*. Your article has been favorably evaluated by Gary Westbrook (Senior editor) and four reviewers, one of whom, Klaus-Armin Nave (Reviewer #1), is a member of our Board of Reviewing Editors. The following individuals involved in review of your submission has agreed to reveal their identity: Klaus-Armin Nave (Reviewer #1 and Reviewing Editor) and Patrizia Casaccia (Reviewer #2). The reviewers have discussed the reviews with one another and the Reviewing Editor has drafted this decision to help you prepare a revised submission.

Summary:

This manuscript addresses a very important and critical question in myelin biology: the mechanism of remyelination. The manuscript convincingly demonstrates that a very rapid remyelination of spinal cord lesions in EAE mice reduces the extent of neurodegeneration and clinical symptoms. At a general level, the data also suggest that axon loss in EAE is caused by targeted oligodendrocyte/myelin injury (subject to repair) and not by the mere recruitment and presence of highly activated immune cells (bystander effect). Specifically, the paper analyzes the effect of clemastine as a pharmacological inhibitor of muscarinic receptor M1R. The authors show that inhibition or loss of this receptor from oligodendrocyte lineage cells drives remyelination and ameliorates the EAE phenotype.

Essential revisions:

1) The text needs major revisions. The writing style does not match the elegance of the experimental results and it should be edited in its entirety. The use of imprecise terminology, grammatical errors and lack of flow in the overall description of the results negatively impacts reading of the manuscript. The text requires serious editing from Abstract to the end of the Results, as the scientific message is often lost due to difficulties in the flow.

2) The authors' major conclusion that remyelination supports axonal integrity and neuronal function is by itself not so surprising and should be rephrased in the Abstract. More remarkable is the unexpected early onset of remyelination that overlaps and interferes with the peak of EAE scores. As the authors point out, it demonstrates that myelin itself is axon protective in neuroinflammation. This could be made clearer in the Abstract.

3) Demonstrating that Clemastine does not interfere with immunization is critical and is so far done indirectly in conditional M1R mutants. Which immune cells express muscarinic receptors? Papers reporting immunosuppressive effects of this drug should be discussed (e.g. Jutel et al., Nature 2001, PMID: 11574888; Johansen et al. J Allergy Clin Immunol. 2011 PMID: 21807405). However, the deletion of M1R in OPC may theoretically also influence the production of substances from OPC with the ability to modulate immune cell responses. Thus in some of their experiments the reduced axonal damage which is associated with remyelination and an improved clinical score may also be positively influenced by an altered inflammatory milieu. The authors should stain for activated microglia and demonstrate similar T cell counts and/or cytokine expression levels in MOG-EAE mice with and without Clemastine treatment. Also in Figure 5, an important control is to show the level of immune infiltration in WT and M1RKO and compare it to the conditional M1RKO in Figure 7. If some of that is beyond the scope, the authors must be more cautious in their interpretation and discussion of their results.

4) In the Abstract and elsewhere states the necessity of ".. uncoupling the immunological response and degenerative processes from possible consequent and even coincident repair…" – it is unclear that these processes have been "uncoupled", even when using OPC-specific Cre mice. In fact, with the overlap in time between reduced EAE scores and remyelination all processes appear even more interrelated than if remyelination were following several days after EAE peak scores.

5) The quality of immunohistochemistry for NF and MOG in Figure 1 is suboptimal and should be replaced or omitted. High resolution and quality images of longitudinal spinal cord section, stained for NF and MOG from clemastine treated mice should be included in Figure 2. The effect of clemastine treatment on overall infiltration of immune cells in the spinal cord could be presented in the same figure.

6) The authors may include the efficiency of recombination for the NG2-CreErt;Tau-mGFP line and explain why only "mature" oligodendrocyte express Tau (shouldn't also progenitor and pre-myelinating oligodendrocyte express it?). Figure 3 does not convey the most important message and it is best if combined with the data currently in supplemental. Quantification of the "tau" positive rings in untreated and clemastine treated mice would provide strong support to the remyelinating effect of the drug.

7) In the Introduction authors state: "…the precise mechanisms underlying axonal degeneration remains unknown (Bjartmar and Trapp, 2003)". This may still be broadly true but several ideas have accumulated over the past decade or so since the publication of this review. They should cite more recent literature on this topic.

---

## [Author Response]

*1) The text needs major revisions. The writing style does not match the elegance of the experimental results and it should be edited in its entirety. The use of imprecise terminology, grammatical errors and lack of flow in the overall description of the results negatively impacts reading of the manuscript. The text requires serious editing from Abstract to the end of the Results, as the scientific message is often lost due to difficulties in the flow.*

We apologize for the writing style, grammatical errors and imprecise terminology. We have now made major revisions to the text.

*2) The authors' major conclusion that remyelination supports axonal integrity and neuronal function is by itself not so surprising and should be rephrased in the Abstract. More remarkable is the unexpected early onset of remyelination that overlaps and interferes with the peak of EAE scores. As the authors point out, it demonstrates that myelin itself is axon protective in neuroinflammation. This could be made clearer in the Abstract.*

We now emphasize, “accelerated remyelination” and have tried to clarify that myelin prevents axon loss after inflammatory demyelination in the Abstract.

*3) Demonstrating that Clemastine does not interfere with immunization is critical and is so far done indirectly in conditional M1R mutants. Which immune cells express muscarinic receptors? Papers reporting immunosuppressive effects of this drug should be discussed (e.g. Jutel et al., Nature 2001, PMID: 11574888; Johansen et al. J Allergy Clin Immunol. 2011 PMID: 21807405). However, the deletion of M1R in OPC may theoretically also influence the production of substances from OPC with the ability to modulate immune cell responses. Thus in some of their experiments the reduced axonal damage which is associated with remyelination and an improved clinical score may also be positively influenced by an altered inflammatory milieu. The authors should stain for activated microglia and demonstrate similar T cell counts and/or cytokine expression levels in MOG-EAE mice with and without Clemastine treatment. Also in Figure 5, an important control is to show the level of immune infiltration in WT and M1RKO and compare it to the conditional M1RKO in Figure 7. If some of that is beyond the scope, the authors must be more cautious in their interpretation and discussion of their results.*

We agree with the reviewers that it is impossible to be completely confident that small molecule therapeutics for remyelination (such as clemastine) will not also affect many other processes. With that said, we have now extensively discussed and referenced this possibility, as well as the potential effects that OPCs may have on the immune response. Additionally, we have now examined T-cell and macrophage infiltration and microglial activation in EAE lesions after clemastine treatment (Figure 2) and in the Chrm1 cKO mice (Figure 7—figure supplement 3). We do not find any differences in the number of CD3 or Iba1 positive cells within the lesions after clemastine treatment or in the Chrm1 cKO mice.

Additionally, to examine the possibility that Chrm1 deletion in cells other than oligodendroglia contributes to the prevention of axon loss, we assessed unmyelinated axons in the spinal cord, which were exposed to a similar inflammatory milieu as the myelinated axons. CGRP positive unmyelinated axons in the dorsal horn of the spinal cord were similar in density between the naïve wildtype and Chrm1 knockout mice (Figure 5). When subjected to EAE, the density of CGRP positive fibers was dramatically decreased in both Chrm1 null and wildtype mice (Figure 5). It is noteworthy that the decrease in the CGRP positive fibers is similar between wildtype and Chrm1 knockout mice, indicating that Chrm1 deletion does not rescue unmyelinated axons from degeneration and is supportive of the possibility that accelerated remyelination protects axons from degeneration after EAE.

*4) In the Abstract and elsewhere states the necessity of ".. uncoupling the immunological response and degenerative processes from possible consequent and even coincident repair…" – it is unclear that these processes have been "uncoupled", even when using OPC-specific Cre mice. In fact, with the overlap in time between reduced EAE scores and remyelination all processes appear even more interrelated than if remyelination were following several days after EAE peak scores.*

We agree that all of these processes seem temporally and spatially interrelated—however, the goal of our study was to uncouple these processes mechanistically by taking a genetic approach to manipulate oligodendroglial function without altering inflammation or degeneration. We have now added the statement that remyelination is “coincident” with inflammation and the degenerative process in the abstract and throughout the manuscript.

*5) The quality of immunohistochemistry for NF and MOG in Figure 1 is suboptimal and should be replaced or omitted. High resolution and quality images of longitudinal spinal cord section, stained for NF and MOG from clemastine treated mice should be included in Figure 2. The effect of clemastine treatment on overall infiltration of immune cells in the spinal cord could be presented in the same figure.*

We have now replaced the longitudinal immunostaining of NF and MOG in Figure 1 and replaced it with high-resolution spinal cord cross sections. Additionally, we have also added the quantification of the density of T-cell and macrophage infiltration and microglial activation in EAE lesions after clemastine treatment to Figure 2.

*6) The authors may include the efficiency of recombination for the NG2-CreErt;Tau-mGFP line and explain why only "mature" oligodendrocyte express Tau (shouldn't also progenitor and pre-myelinating oligodendrocyte express it?). Figure 3 does not convey the most important message and it is best if combined with the data currently in supplemental. Quantification of the "tau" positive rings in untreated and clemastine treated mice would provide strong support to the remyelinating effect of the drug.*

We have now examined the recombination efficiency of the *Cspg4*-CreErt; *Mapt*-mGFP line. As newly formed oligodendrocytes are not generated in high frequency in adult mice, we examined 14-day old mice to determine recombination efficiency by administration of tamoxifen at postnatal day 8 and then examining recombination 6 days later. By examining cortical regions where myelination is sparse at postnatal day 14, we quantified the number of GFP positive, CC1 positive oligodendrocytes and calculated a recombination efficiency of approximately 30% of the total number of oligodendrocytes (Figure 3—figure supplement 1). It is also important to note that we did not identify GFP positive cells in the cortex that were not CC1 positive, suggesting that GFP expression was specific to oligodendrocytes. Taken together, while this approach will identify newly generated oligodendrocytes and remyelination in EAE, it does not necessarily allow us to identify all of the remyelinated axons. We have now discussed this in detail in the manuscript and added an additional reference for the mouse line that was used to identify newly differentiated oligodendrocytes during development (Etxeberria et al., 2016).

Quantification of GFP positive myelin was difficult to evaluate, as we did not detect any GFP positive myelin sheaths in the vehicle control group when compared to the clemastine treated mice. We have now discussed this in the last paragraph of the subsection “Small molecule approaches do not definitively demonstrate remyelination in EAE”. Our main point in using this mouse line was an attempt to identify “newly” formed remyelinated axons at the peak of EAE – which we were successful in demonstrating.

*7) In the Introduction authors state: "…the precise mechanisms underlying axonal degeneration remains unknown (Bjartmar and Trapp, 2003)". This may still be broadly true but several ideas have accumulated over the past decade or so since the publication of this review. They should cite more recent literature on this topic.*

We have now added additional references (Simons et al., 2014; Singh et al., 2013; Sorbara et al., 2014) in the Introduction concerning degeneration of axons in MS.